# Cytoplasmic RRM1 activation as an acute response to gemcitabine treatment is involved in drug resistance of pancreatic cancer cells

**Tomotaka Kato**[1ᵒ], **Hiroaki Ono**[1ᵒ]*, **Mikiya Fujii**[1], **Keiichi Akahoshi**[1], **Toshiro Ogura**[1], **Kosuke Ogawa**[1], **Daisuke Ban**[1], **Atsushi Kudo**[1], **Shinji Tanaka**[2], **Minoru Tanabe**[1]

1 Department of Hepatobiliary and Pancreatic Surgery, Graduate School of Medicine, Tokyo Medical and Dental University, Tokyo, Japan, 2 Department of Molecular Oncology, Graduate School of Medicine, Tokyo Medical and Dental University, Tokyo, Japan

ᵒ These authors contributed equally to this work.

* ono.msrg@tmd.ac.jp

**Data Availability Statement:** All relevant data are within the paper and its Supporting Information files.

## Abstract

### Background

RRM1 is functionally associated with DNA replication and DNA damage repair. However, the biological activity of RRM1 in pancreatic cancer remains undetermined.

### Methods

To determine relationships between RRM1 expression and the prognosis of pancreatic cancer, and to explore RRM1 function in cancer biology, we investigated RRM1 expression levels in 121 pancreatic cancer patients by immunohistochemical staining and performed in vitro experiments to analyze the functional consequences of RRM1 expression.

### Results

Patients with high RRM1 expression had significantly poorer clinical outcomes (overall survival; $p = 0.006$, disease-free survival; $p = 0.0491$). In particular, high RRM1 expression was also associated with poorer overall survival on adjuvant chemotherapy ($p = 0.008$). We found that RRM1 expression was increased 24 hours after exposure to gemcitabine and could be suppressed by histone acetyltransferase inhibition. RRM1 activation in response to gemcitabine exposure was induced mainly in the cytoplasm and cytoplasmic RRM1 activation was related to cancer cell viability. In contrast, cancer cells lacking cytoplasmic RRM1 activation were confirmed to show severe DNA damage. RRM1 inhibition with specific siRNA or hydroxyurea enhanced the cytotoxic effects of gemcitabine for pancreatic cancer cells.

**Funding:** This work was supported by JSHPS KAKENHI for Grant-Aid for Young Scientists 18K15267. The funders had no role in study design, data collection and analysis, decision to publish, or preparation of the manuscript.

**Competing interests:** The authors have declared that no competing interests exist.

## Conclusions

Cytoplasmic RRM1 activation is involved in biological processes related to drug resistance in response to gemcitabine exposure and could be a potential target for pancreatic cancer treatment.

## Introduction

Pancreatic cancer is commonly progressive and fatal. Despite improvements in diagnosis and chemotherapy, its incidence and mortality have been increasing, such that it is now the fourth leading cause of cancer death in the United States [1–3]. In general, the highly malignant properties of pancreatic cancer are due to its aggressive biological behavior and early acquisition of drug resistance [4,5]. Recently, however, several new regimens have been established which may have marked therapeutic effects [6–8]. Combination therapies play a central role in treatment strategies for patients with pancreatic cancer. Nevertheless, there are many undetermined molecular mechanisms related to the acquisition of pancreatic cancer drug resistance.

Gemcitabine is currently a key agent for the treatment of pancreatic cancer. Once this drug is taken up into cells through the membrane nucleoside transporter hENT1, it is phosphorylated by deoxycytidine kinase (dCK) to generate gemcitabine monophosphate (dFdCMP) and then subsequently converted to gemcitabine diphosphate (dFdCDP) and finally gemcitabine triphosphate (dFdCTP), which is the active form [9,10]. Incorporation of dFdCTP into tumor DNA is the fundamental mode of action responsible for gemcitabine cytotoxicity [9,11]. However, dFdCDP-induced inhibition of ribonucleotide reductase (RR) is a different mechanism enhancing gemcitabine cytotoxicity [9,11,12]. Ribonucleotide reductase is mainly responsible for controlling DNA biosynthesis through the formation of dNTPs [9,13] and is also essential for DNA damage repair to aid recovery from cytotoxic agents including gemcitabine. Pharmacologically, gemcitabine reduces the activity of RR, which in turn reduces the endogenous dNTP pool, thus indirectly facilitating dFdCTP incorporation into DNA [9,11].

Ribonucleotide reductase large subunit M1 (RRM1) is a rate-limiting enzyme of the DNA synthesis pathway, and is depended on for the conversion of ribonucleotides to dNTPs [14]. Studies have shown that increased levels of RRM1 are associated with resistance to gemcitabine [15,16]. RRM1 has also been reported to affect disease prognosis of pancreatic cancer. In fact, the meta-analysis for RRM1 demonstrated that high expression of RRM1 was significantly associated with worse overall survival. The clinical impact of RRM1 has been elucidated [17–22]. However, the functions of RRM1 in pancreatic cancer biology remain unclear.

In the present study, we investigated RRM1 expression levels in pancreatic cancer by immunohistochemical staining of tumor sections and attempted to determine correlations with clinicopathological factors and survival. We also carried out in vitro experiments to analyze the functional consequences of RRM1 expression, especially focusing on DNA damage in pancreatic cancer cells in response to gemcitabine exposure. Our study provides evidence that RRM1 could be an effective target in pancreatic cancer treatment.

## Methods

### Materials

Culture media (DMEM), fetal bovine serum (FBS) and penicillin/streptomycin (P/S) were obtained from Sigma-Aldrich (St. Louis, MO). Anti-RRM1 (D12F12; #8637), anti-Phospho-

Histone H2A.X (Ser139, D7T2V; #80312S), anti-GAPDH (D16H11; #5174), anti-Acetyl Histone H3 (Lys9) (C5B11; #9649S), anti-Acetyl Histone H3 (Lys27) (D5E4; #8173S), and anti-Histone H3 (D1H2; #4499) antibodies were from Cell Signaling Technology (Danvers, MA). Anti-Lamin B antibody (M-20; sc-6217) was obtained from Santa Cruz Biochemistry (Santa Cruz, CA). Anti-$\alpha$-Tubulin antibody (DM1A; T9026) was from Sigma-Aldrich (St. Louis, MO). Gemcitabine hydrochloride (G6423), the histone acetyltransferase inhibitor C646 (SML0002) and the ribonucleotide reductase (RNR) inhibitor Hydroxyurea (H8627) were all from Sigma-Aldrich.

## Cell cultures

The human pancreatic cancer cell lines Hs766T, MIAPaCa2, Panc1 and PSN1 were obtained from the American Type Culture Collection (ATCC, Manassas, VA). The MIAPaCa2, Panc1, and PSN1 lines were obtained in August 2016 and Hs766T in July 2019. All cancer cell lines were authenticated using STR analysis for DNA profiling and all experiments were performed with mycoplasma-free cells. Cancer cells were maintained in high glucose DMEM medium containing 10% FBS and 1% P/S in a humidified 5% $CO_2$ chamber at 37˚C.

## Human tissues (for IHC)

Archived tissue slides were obtained from the Department of Pathology at Tokyo Medical and Dental University Hospital, Tokyo, Japan, for the patients with pancreatic cancer who underwent radical operation from February 2005 to April 2017. With approval of the ethics committees of the Faculty of Medicine in Tokyo Medical and Dental University (permission No. M2000-1080, G2017-018), written informed consent to have data from their medical records used in research was obtained from all patients. Patients were anonymously coded in accordance with ethical guidelines, as instructed by the Declaration of Helsinki.

Archived tissues of pancreatic ductal adenocarcinoma from 121 different patients and low-malignant pancreatic disease from 12 different patients were included in this study. Heat-induced epitope retrieval was performed with citrate buffer (pH 6.0). Anti-RRM1 antibody was diluted 1/200 with SignalStain Antibody Diluent (#8112; Cell Signaling Technology) and incubated for 1 hour at room temperature. Antigen-antibody reactions were visualized with SignalStain Boost IHC Detection Reagents (HRP, Rabbit #8114; Cell Signaling Technology).

## Western blotting

Western blotting was performed as previously described [23]. GAPDH and Histone H3 served as a loading control for the normalization of each lane. Protein bands were visualized using ImageQuant LAS 4000 mini (GE Healthcare, Chicago, IL). Western blotting was repeated at least three times with similar results and representative blots are presented. Densitometric analysis was conducted using ImageQuant TL (GE Healthcare) to calculate the intensity of each protein band.

## Gene silencing by small interfering RNA

Loss-of-function analysis was performed using siRNAs targeting RRM1 (HSS109388, Invitrogen, St Louis, MO: sense 5′–CCCAGUUACUGAAUAAGCAGAUCUU–3′, antisense 5′–AAGAU CUGCUUAUUCAGUAACUGGG–3′) and negative control (Stealth RNAiTM Negative Control Med GC Duplex #2, Invitrogen). An alternative sequence of siRNA targeting RRM1 (HSS184469, Invitrogen: sense 5′–CAGAAGCUUUGUUAUGGACUCAAUA–3′, antisense-5′–UAUUGAGUCCAUAACAAAGCUUCUG–  3′) was also used with similar results. Each siRNA

(10–20 nmol/l) was transfected into pancreatic cancer cells using Lipofectamine RNA iMAX (Invitrogen) according to the manufacturer's instructions. Knockdown of each target gene was confirmed by Western blotting.

## Cell viability assay

Cell numbers were evaluated by WST-8 assay (Cell Counting Kit-8; Dojindo Molecular Technologies, Gaithersburg, MD), as previously described [23]. In brief, 5.0–7.5 x $10^3$ cells per well were seeded into 96-well plates and incubated overnight at 37˚C. Following 72 hours of gemcitabine treatment, the cell viability assay was performed according to the manufacturer's instructions. The absorbance of each well was measured at 450 nm using an iMarkTM microplate reader (Bio-Rad Laboratories) and was within the linear range of the assay.

## Quantitative real-time RT–PCR

Extracted RNA was reverse transcribed into first-strand cDNA using SuperScript VILO cDNA Synthesis kits (Invitrogen, Carlsbad, CA). Expression of mRNA was determined using Taq-Man Gene Expression Assays (Applied Biosystems, Foster City, CA). The TaqMan assays used in this study were RRM1; Hs01040698_m1. Gene expression values are presented as ratios between genes of interest and an internal reference gene (Hs99999901_s1 for eukaryotic 18S), and subsequently normalized against the value for the control (relative expression level). Normal pancreas RNA (Human Adult Normal Tissue 5 Donor Pool: #R1234188-P) was purchased from BioChain Institution (Newark, CA). Each assay was performed in duplicate for each sample.

## Immunofluorescence staining

Cells were fixed in 4% formaldehyde diluted in phosphate-buffered saline (PBS) for 15 minutes, permeabilized with 0.2% Triton X-100, treated with blocking buffer (1% BSA in PBS), and then incubated overnight with the primary antibody at 4˚C. Cells were then incubated with the secondary antibody (anti-mouse Alexa Fluor 488 (#4408) or anti-rabbit Alexa Fluor 555 (#4413), Cell Signaling Technologies) for 1 hour at room temperature.

## Cytoplasmic and nuclear fractions

Whole cell lysates were further fractionated into subcellular components using the NE-PER™ cytoplasm/nuclear protein extraction kit (Thermo Fisher Scientific), according to the manufacturer's instructions. For each fractionation, the protein concentration was determined by BCA Protein Assay (Pierce Biotechnology). 10–20 μg of protein from each extract were resolved by 4–20% SDS-PAGE and evaluated by Western blotting.

## Statistical analysis

Clinicopathological factors were compared using Mann-Whitney U testing and the Chi-squared test. Multivariate analysis used a logistic regression model. The cumulative survival rate was estimated by the Kaplan-Meier method, and significance was determined using the log-rank test and the Gehan-Breslow-Wilcoxon test. Experiments were conducted in triplicate in independent settings and the values presented represent their means using the Student's t-test for single comparisons. P <0.05 was defined as statistically significant. Statistical analyses were performed using SPSS for Windows, version 25.0 (SPSS Inc., Chicago, IL) and GraphPad Prism 7 software (GraphPad Software Inc., San Diego, CA). Drawing figures and IC 50 calculation were done with GraphPad Prism 7 software.

## Results

### High RRM1 expression is associated with poorer prognosis in pancreatic cancer in a disease-specific manner

In the present study, we examined RRM1 expression and determined its clinical and biological significance in pancreatic cancers. First, RRM1 expression in pancreatic cancer tissues of 121 patients who underwent surgical resection was determined by immunohistochemical staining (Fig 1A). This showed that RRM1 was overexpressed in cancer cells but not in adjacent normal pancreatic tissues. RRM1 expression status was classified into low and high expression groups (Fig 1B). RRM1 expression was also examined in different types of low-grade malignant pancreatic diseases such as intraductal papillary mucinous neoplasia (IPMN) with low and intermediate-grade dysplasia, serous cyst neoplasia (SCN), and mucinous cyst neoplasia (MCN) (Figs 1C and S1). Notably, no RRM1 expression was observed in 12 samples of tumor tissues of these benign diseases, suggesting that its overexpression may be specific for malignant cancer.

The clinical significance of RRM1 expression in pancreatic cancer cells was then evaluated. The median follow-up for the patients in this study was 22.6 months. RRM1 overexpression was observed in 85 patients (70.2%). As shown in Fig 1D, overall survival (OS) was significantly worse in patients with high RRM1 expression relative to low RRM1 expression (median OS 21.2 months versus 64.7 months, log-rank test P = 0.006). Disease-free survival (DFS) was also worse for patients whose tumors had high RRM1 expression (median DFS 11.7 months versus 29.0 months, Gehan-Breslow-Wilcoxon test P = 0.0491).

Clinicopathological characteristics, patient demographics, surgical procedure, pathological tumor stage, and adjuvant therapy are depicted in Table 1. There was a significant relationship between RRM1 expression and higher venous invasion. Univariate regression analyses of OS indicated that high CA19-9 level, extrapancreatic invasion, residual tumor, lymphatic invasion, T/N status, non-adjuvant chemotherapy, and RRM expression status were all significant risk factors for pancreatic cancer patients. Multiple regression analysis for these seven variables showed that extrapancreatic invasion [P = 0.002; confidence interval (CI), 1.77–13.9], lymph node metastasis (P <0.001; CI, 1.87–6.42), non-adjuvant chemotherapy (P = 0.005; CI, 1.23–3.29), and high RRM1 (P = 0.017; CI, 1.12–3.23) were independent prognostic factors (Table 2).

We further investigated the relevance of RRM1 expression for postoperative prognosis focusing on the effects of adjuvant chemotherapy (Fig 2A and 2B). In this study, 79 patients (65%) received adjuvant chemotherapy with gemcitabine or S-1 after radical surgery; 29 patients received gemcitabine (GEM group) and 50 patients with S-1 (S-1 group). 42 patients received no adjuvant chemotherapy (No adjuvant group). As shown in Fig 2A, patients with pancreatic cancers expressing high levels of RRM1 had poorer OS following adjuvant chemotherapy.

In addition, we compared the therapeutic efficacy of adjuvant chemotherapy in low and high RRM1 expression. In pancreatic cancer with high RRM1, there was little therapeutic effect of adjuvant chemotherapy (Fig 2C). On the other hand, in pancreatic cancer with low RRM1, there was a trend toward a benefit from additional postoperative adjuvant chemotherapy comparing to no adjuvant group, although the difference was not statistically significant (Fig 2D). These results suggested that RRM1 expression status may have an important role in pancreatic cancer chemoresistance in general.

### RRM1 inhibition augments gemcitabine sensitivity of pancreatic cancer cells

Our immunohistochemical analysis indicated the clinical importance of high RRM1 expression in gemcitabine chemoresistance against pancreatic cancer consistent with reported

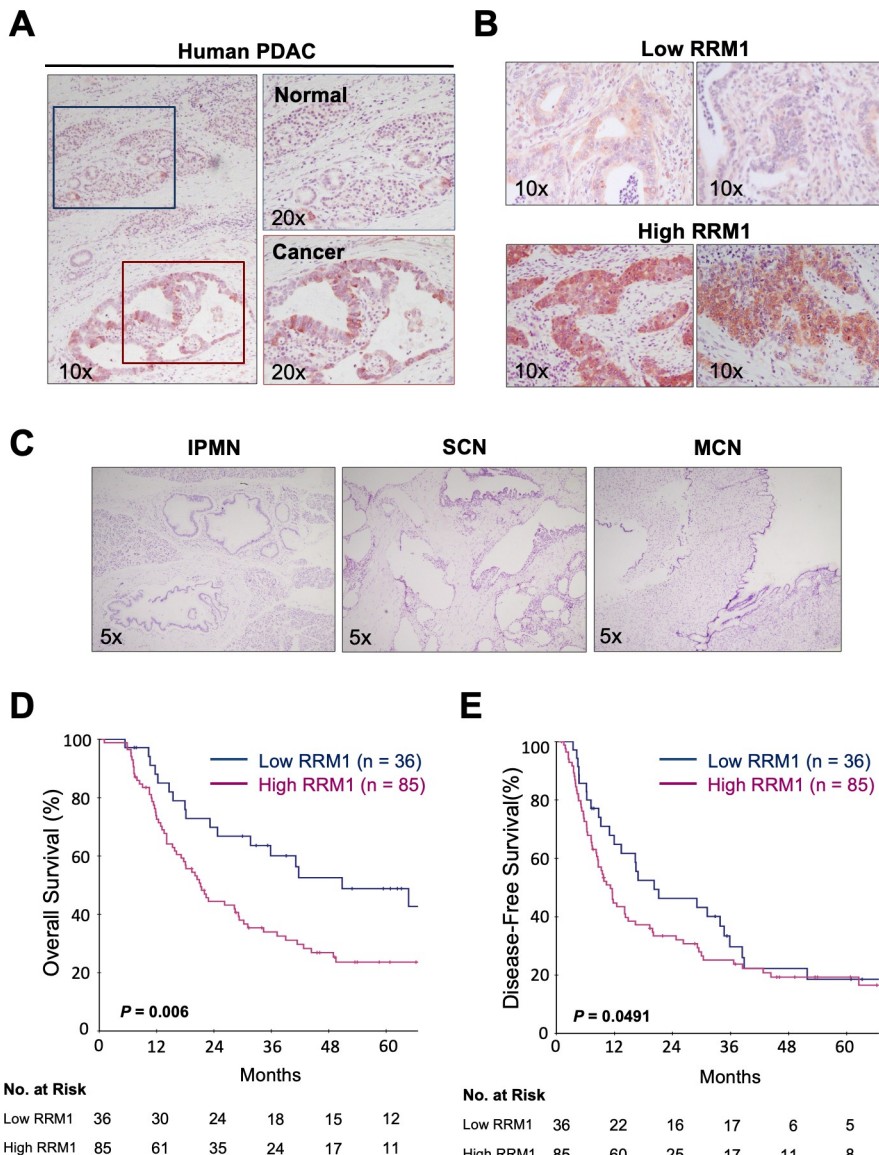

**Fig 1. Association between RRM1 expression and prognosis in pancreatic cancer.** (**A**) Left panel: representative immunohistochemical staining of pancreatic ductal adenocarcinomas with positive RRM1 expression. Right upper panel: adjacent normal pancreatic tissue. Right lower panel: ductal adenocarcinoma. (**B**) Representative immunohistochemical staining of low and high RRM1 expression levels. For the immunohistochemical staining of RRM1 in pancreatic cancer, the islets of Langerhans served as a positive internal control to evaluate the expression. RRM1 expression level was graded as low (no staining or weak intensity staining comparing to internal control in less than 30% of cells) or high (strong intensity staining in more than 30% of cells based on cytoplasmic staining intensity). The staining grade of RRM1 expression was assessed by two investigators and reviewed by one pathologist. (**C**) Immunohistochemical staining of non-malignant pancreatic disease tissues. Left panel, intraductal papillary mucinous neoplasm (IPMN); middle panel, serous cyst neoplasm (SCN); right panel, mucinous cyst neoplasm (MCN). (**D**), (**E**) Kaplan-Meier curves for overall survival (**D**) and disease-free survival (**E**) of patients with pancreatic cancer with high or low RRM1 expression. RRM1 immunoreactivity in tumor cells was significantly associated with a worse prognosis (overall survival, P = 0.006, log-rank test; disease-free survival, P = 0.0491, Gehan-Breslow-Wilcoxon test).

evidence [16,17,24]. However, biological importance of RRM1 expression on gemcitabine resistance remained still uncertain. Therefore, in vitro experiments were carried out to determine the biological and functional effects of RRM1 expression on gemcitabine treatment.

**Table 1. Patient characteristics.**

| | Low RRM1 n = 36 | High RRM1 n = 85 | *P*—value |
|---|---|---|---|
| **Age, year (mean ± SD)** | 68 ± 10.0 | 68 ± 8.7 | 0.823 |
| **Sex** | | | **0.046** |
| Male | 18 (50%) | 58 (68%) | |
| Female | 18 (50%) | 27 (32%) | |
| **CEA, ng/ml (mean ± SD)** | 4.0 ± 10.4 | 7.1 ± 14.8 | 0.228 |
| **CA19-9, U/ml (mean ± SD)** | 427 ± 1134 | 463 ± 1349 | 0.890 |
| **Operation Procedure** | | | 0.413 |
| PD | 24 (67%) | 53 (64%) | |
| DP | 12 (33%) | 28 (33%) | |
| TP | 0 | 4 (3%) | |
| **Extrapancreatic Invasion** | | | 0.103 |
| negative | 9 (25%) | 11 (13%) | |
| positive | 27 (75%) | 74 (87%) | |
| **Portal Vein invasion** | | | 0.99 |
| negative | 28 (78%) | 66 (78%) | |
| positive | 8 (22%) | 19 (22%) | |
| **Venous invasion** | | | **0.044** |
| negative | 12 (11%) | 3 (2%) | |
| positive | 23 (89%) | 82 (98%) | |
| **Neural invasion** | | | 0.855 |
| negative | 2 (6%) | 4(5%) | |
| positive | 34 (94%) | 80 (95%) | |
| **Lymphatic invasion** | | | 0.736 |
| negative | 13 (36%) | 28 (33%) | |
| positive | 23 (64%) | 57 (67%) | |
| **T status (AJCC7)** | | | 0.683 |
| pT1, < 20mm | 7 (19%) | 11 (13%) | |
| pT2, 20mm - 40mm | 19 (53%) | 50 (59%) | |
| pT3, 40mm ≤ | 10 (28%) | 24 (28%) | |
| **N status (AJCC7)** | | | 0.827 |
| pN0 | 11 (31%) | 25 (29%) | |
| pN1 | 25 (69%) | 60 (71%) | |

Bold represents P-value <0.05, CEA: Carcinoembryonic antigen, CA19-9: Carbohydrate antigen 19–9, PD; Pancreatoduodenectomy, DP: Distal pancreatectomy, TP: Total pancreatectomy, AJCC7: American Joint Commission Cancer 7th edition.

We quantified RRM1 in four established human pancreatic cancer cell lines (Hs766T, MIA-PaCa2, PSN1, and Panc1). Real time RT-PCR analysis demonstrated that RRM1 mRNA expression was aberrantly activated in all pancreatic cancer cell lines comparing to normal pancreas tissue. And RRM1 protein expression was confirmed at various levels by Western blotting in the panel of pancreatic cancer cell lines (Fig 3A).

We next analyzed the effect on cell viability of reducing RRM1 function after gemcitabine exposure using specific RRM1 siRNAs or hydroxyurea, the latter being a clinically-used agent that inhibits ribonucleotide reductase [25]. Notably, RRM1-specific siRNA treatment downregulated cell viability of pancreatic cancer cells. Furthermore, cytotoxic effects after gemcitabine exposure were significantly increased in pancreatic cancer cells on treatment with RRM1-specific siRNA, relative to controls in both MIAPaCa2 and PSN1 cells (Fig 3B). Western blotting

**Table 2. Cox proportional analysis for the prognostic factors of overall survival.**

| Characteristics | Category | Univariate analysis | | Multivariate analysis | |
|---|---|---|---|---|---|
| | | Hazard (95% CI) | *P*—value | Hazard (95% CI) | *P*—value |
| Age | >75 | 1.38 (0.86–2.20) | - | | |
| Sex | male | 1.19 (0.77–1.84) | - | | |
| CEA, ng/ml | >5 | 1.57 (0.99–2.48) | - | | |
| CA19-9, U/ml | >100 | 1.60 (1.04–2.45) | 0.031 | - | - |
| Surgical Procedure | PD | 1.42 (0.92–2.18) | - | | |
| Extrapancreatic Invasion | positive | 5.19 (2.09–12.9) | < 0.001 | 4.96 (1.77–13.9) | 0.002 |
| Residual Tumor | positive | 1.56 (1.02–2.40) | 0.042 | - | - |
| Venous Invasion | positive | 4.53 (0.63–32.6) | - | | |
| Neural Invasion | positive | 4.07 (0.99–16.8) | - | | |
| Lymphatic Invasion | positive | 2.76 (1.62–4.69) | < 0.001 | - | - |
| T status (AJCC7) | ≥ T3 | 1.79 (1.15–2.80) | 0.011 | - | - |
| N status (AJCC7) | ≥ N1 | 3.64 (2.01–6.59) | < 0.001 | 3.46 (1.87–6.42) | < 0.001 |
| Adjuvant chemotherapy | none | 1.61 (1.03–2.53) | 0.036 | 2.01 (1.23–3.29) | 0.005 |
| RRM1 | high | 2.00 (1.21–3.31) | 0.004 | 1.90 (1.12–3.23) | 0.017 |

CI: Confidence interval, CEA: Carcinoembryonic antigen, CA19-9: Carbohydrate antigen 19–9, PD; pancreatoduodenectomy, AJCC7: American Joint Commission Cancer 7th edition, RRM1: Ribonucleotide reductase large subunit M1.

analysis in these two cell lines showed that exposure to RRM1 siRNA also increased γ-H2AX expression, a surrogate marker for DNA double-strand breaks (Fig 3D). Additional administration of gemcitabine after RRM1 siRNA exposure potently increased γ-H2AX expression, indicating that RRM1 inhibition contributes to augmentation of gemcitabine sensitivity and enhances DNA damage after exposure to the drug.

Similar effects on gemcitabine-induced cytotoxicity were confirmed using ribonuclease inhibition. Hydroxyurea also inhibited cell viability of pancreatic cancer cells (Fig 3C). In addition, hydroxyurea coadministration together with gemcitabine apparently decreased cancer cell viability and increased DNA damage accumulation in both MIAPaCa2 and PSN1 cancer cells, as shown in Fig 3E. Our results supported the hypothesis that RRM1 inhibition enhances the effect of gemcitabine, which could be useful for the treatment of pancreatic cancer.

## Increased RRM1 expression in pancreatic cancer cell cytoplasm after gemcitabine exposure is related to resistance to the drug

The biological effects of RRM1 expression after gemcitabine exposure were further investigated. We focused on the location of RRM1 expression in pancreatic cancer cells and found that it was mainly present in the cytoplasm, as shown by immunofluorescence and immunohistochemical staining (Fig 4A). Western blotting confirmed the presence of RRM1 in the cytoplasm (Fig 4B).

We found that gemcitabine exposure alters the level of RRM1 expression. The density of RRM1 expression increased and molecular mass changed after gemcitabine exposure observed by 24 hours after treatment initiation (Fig 4C). Of particular note, this alteration was evoked before γ-H2AX activation in MIAPaC2 cells, suggesting that RRM1 is involved in the signaling response following gemcitabine exposure. RRM1 alteration after gemcitabine exposure was then evaluated using 3 pancreatic cancer cell lines (Fig 4D). Band shifted RRM1 alteration after gemcitabine exposure was most prominent in Panc1 cells (Fig 4C and 4D). Among pancreatic cancer cell lines we tested, Panc1 cells exhibited the most resistant phenotype to

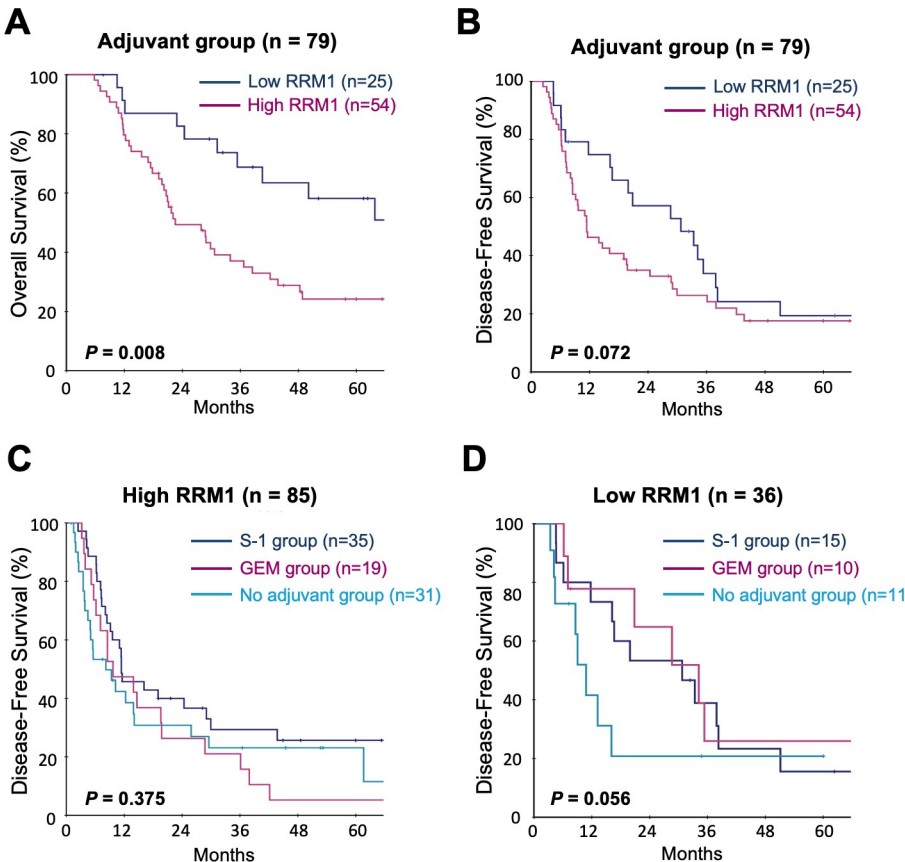

**Fig 2. Correlation between the level of expression of RRM1 and postoperative outcomes in patients who received adjuvant chemotherapy.** Kaplan-Meier curve for overall survival (**A**) and disease-free survival (**B**) of 79 patients who received adjuvant chemotherapy. Kaplan-Meier curve for disease-free survival of 85 patients with high RRM1 expression (**C**) and 36 patients with low RRM1 expression (**D**) in the comparison of receiving adjuvant chemotherapy and no adjuvant chemotherapy. S-1 group: patients who received S-1 adjuvant chemotherapy. GEM group: patients who received gemcitabine adjuvant chemotherapy. No adjuvant group: patients who received no adjuvant chemotherapy, respectively.

gemcitabine (S2B Fig). Furthermore, RRM1 alteration after gemcitabine exposure was significantly scant in PSN1 cells (S3 Fig), which are gemcitabine-sensitive in pancreatic cancer cells lines. However, basal expression of RRM1 in PSN1 cells was higher than MIAPaCa2 cells (Fig 3A). It is likely that the band shifted RRM1 alteration is related to DNA damage repair response rather than the baseline level of RRM1 expression.

This band shift of RRM1 expression may be due to post-transcriptional modification. Histone acetylation is one mechanism that might account for this, and we had previously reported that histone acetylation has a role in early DNA damage repair signaling after gemcitabine exposure in pancreatic cancer cells [26]. Therefore, we attempted to determine the effect of histone acetylation on the alteration of RRM1 expression. To this end, the histone acetyltransferase inhibitor C646 was tested, and as shown in Fig 4E. Notably, original RRM1 expression was inhibited by C646 treatment. Furthermore, band shifted RRM1 expression was also prevented by this agent. Histone acetylation might therefore be associated with both RRM1 expression and the DNA damage response after gemcitabine exposure.

We also found that RRM1 expression was mainly increased in the cytoplasm of cancer cells (Fig 5A); similarly, immunofluorescence staining revealed that RRM1 expression was activated

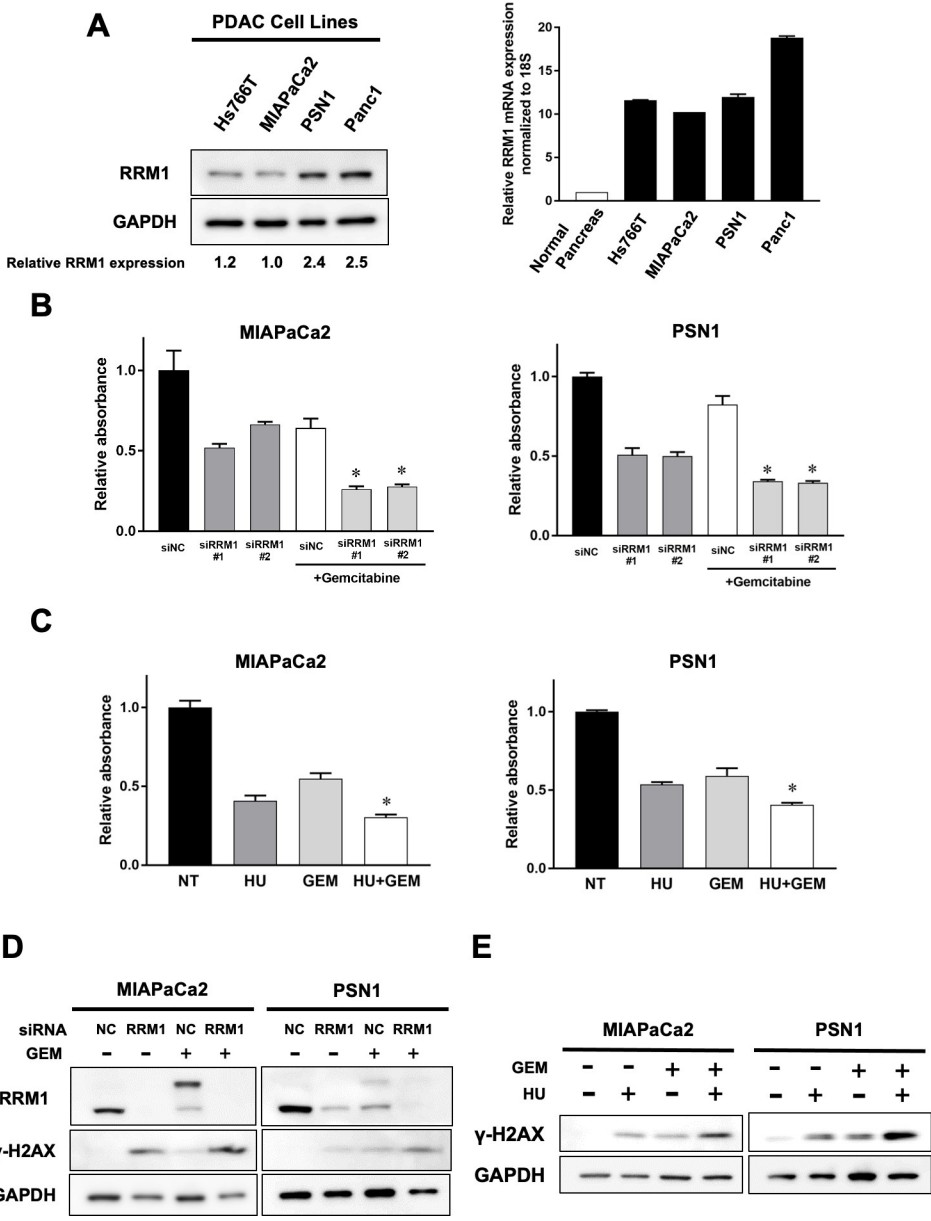

**Fig 3. Augment of gemcitabine sensitivity of pancreatic cancer cells associated with RRM1 inhibition. (A)** Left panel: Endogenous RRM1 protein expression in human pancreatic cancer cell lines by Western blotting. The numbers at the bottom indicate the relative intensity of the bands for RRM1 expression of Hs766T, PSN1, and Panc1 comparing to MIAPaCa2, which showed the lowest endogenous RRM1 expression in the panel (normalized by GAPDH). Right panel: Endogenous RRM1 mRNA expression relatively comparing to normal pancreas tissue. Of note, RRM1 expression was activated at various levels in four pancreatic cancer cell lines. **(B)** Cell viability assay after gemcitabine treatment for 72 hours together with siRRM1 or siNC treatment. Prior to gemcitabine treatment, cells were treated with siRRM1 or siNC for 24 hours. And then cells (MIAPaCa2 $5\times 10^3$, PSN1 $7.5\times 10^3$) were seeded and incubated overnight. Attached cells were treated with or without gemcitabine treatment (15 nM for MIAPaCa2 and 5 nM for PSN1 cells) for 72 hours. Error bars represent mean ± SD. $^*p<0.05$ vs siNC-treated cells with gemcitabine. Each data point was evaluated as relative absorbance normalized to siNC control without treatment. Each experiment was performed in duplicate. **(C)** Cell viability assay after hydroxyurea, gemcitabine, and gemcitabine coadministering with hydroxyurea for 72 hours. Drug concentration of hydroxyurea was determined based on the data of IC50 in each cell (S2A Fig). Hydroxyurea was treated at 750 μM for MIAPaCa2 and 300 μM for PSN1 cells. Gemcitabine was treated at 15 nM for MIAPaCa2 and 5 nM for PSN1 cells. Error bars represent mean ± SD. $^*p<0.05$ vs cells with gemcitabine-treated cells. Each data point was evaluated as relative absorbance normalized to no treatment control. Each experiment was performed in duplicate. **(D), (E)** Effects of RRM1 gene-silencing (**D**) and hydroxyurea (**E**) on gemcitabine-induced DNA damage. DNA damage was evaluated by γ-H2AX expression using Western blotting. **(D)**

Cancer cells were treated with siRRM1 or siNC for 6 hours prior to gemcitabine exposure for 72 hours. Gemcitabine was treated at 25 nM for MIAPaCa2 and 10 nM for PSN1 cells. (**E**) Cancer cells were treated with hydroxyurea, gemcitabine, or coadministrating with hydroxyurea and gemcitabine for 72 hours. Gemcitabine was treated at 15 nM for MIAPaCa2 and 5 nM for PSN1 cells and hydroxyurea at 750 μM for MIAPaCa2 and 300 μM for PSN1 cells.

in the cytoplasm. Interestingly, this finding was only observed in those cancer cells without an accumulation of DNA damage (Fig 5B). These cells might have initiated the acquisition of resistance to DNA damage by gemcitabine. Consistent with this, cells undergoing apoptosis with severe DNA damage did not show activation of RRM1 expression in the cytoplasm.

In addition, we investigated by quantitative analysis whether cytoplasmic RRM1 expression was associated with the act of overcoming DNA damage after gemcitabine exposure. Indeed, cytoplasmic RRM1 expression after gemcitabine exposure was activated mainly in attached and viable cancer cells. In contrast, cancer cells free-floating in the medium and undergoing apoptosis increased DNA damage and cytoplasmic RRM1 activation was significantly abolished (Fig 5C). Taken these findings together, our study provided evidence that increased RRM1 expression in the cytoplasm may have an important role in the process of DNA repair signaling early after gemcitabine exposure.

## Discussion

Ribonuclease reductase (RR) plays a pivotal role in DNA damage repair and DNA replication via the generation of dNTP pools. This enzyme is also involved in gemcitabine metabolism. Several studies revealed that levels of RRM1, which is an isoform of ribonuclease reductase, are associated with postoperative outcomes in some types of cancer [20,21,27,28]. In the present study, we provide evidence that aberrant RRM1 expression is associated with poorer postoperative prognosis in pancreatic cancer. We also documented the absence of RRM1 expression in normal pancreatic tissues and benign pancreatic tumors (Figs 1A and 1C and S1). These results identify aberrant RRM1 expression as regulated in a cancer-specific manner in pancreatic cancer cells.

Recently, combination chemotherapies such as gemcitabine with nab-paclitaxel or FOLFIRINOX (fluorouracil, irinotecan, and oxaliplatin) have achieved some progress in improving the therapeutic effectiveness and have become essential for the treatment of advanced stage pancreatic cancer [6,8]. However, frequent adverse effects preclude long-term treatment with current therapies. Further understanding of the mechanism of gemcitabine or fluorouracil action as the key chemotherapeutic agents is vital for reducing adverse effects but not efficacy.

Even radical surgery for resectable pancreatic cancer is insufficient for curative treatment due to the exceptional malignant potential of this tumor. Therefore, adjuvant chemotherapy is required for the improvement of postoperative outcomes. In general, this disease is treated with gemcitabine or S-1, a modified fluoropyrimidine prodrug [13,29,30]. The notable finding from the present study is that RRM1 expression affects postoperative prognosis on adjuvant treatment in pancreatic cancer patients (Fig 2A and 2B). Furthermore, in our study we also found that S-1 showed a statistically significant benefit in postoperative prognosis of patients with low RRM1 expression for the adjuvant chemotherapy (S4A Fig). On the other hand, the difference between patients with pancreatic cancer with low and high RRM1 expression in the GEM group did not reach statistical significance, though there was a trend towards better survival treating with gemcitabine (S4B Fig). The difference of postoperative outcomes may be due to the limited sample size and the different historical background between the GEM and S-1 group.

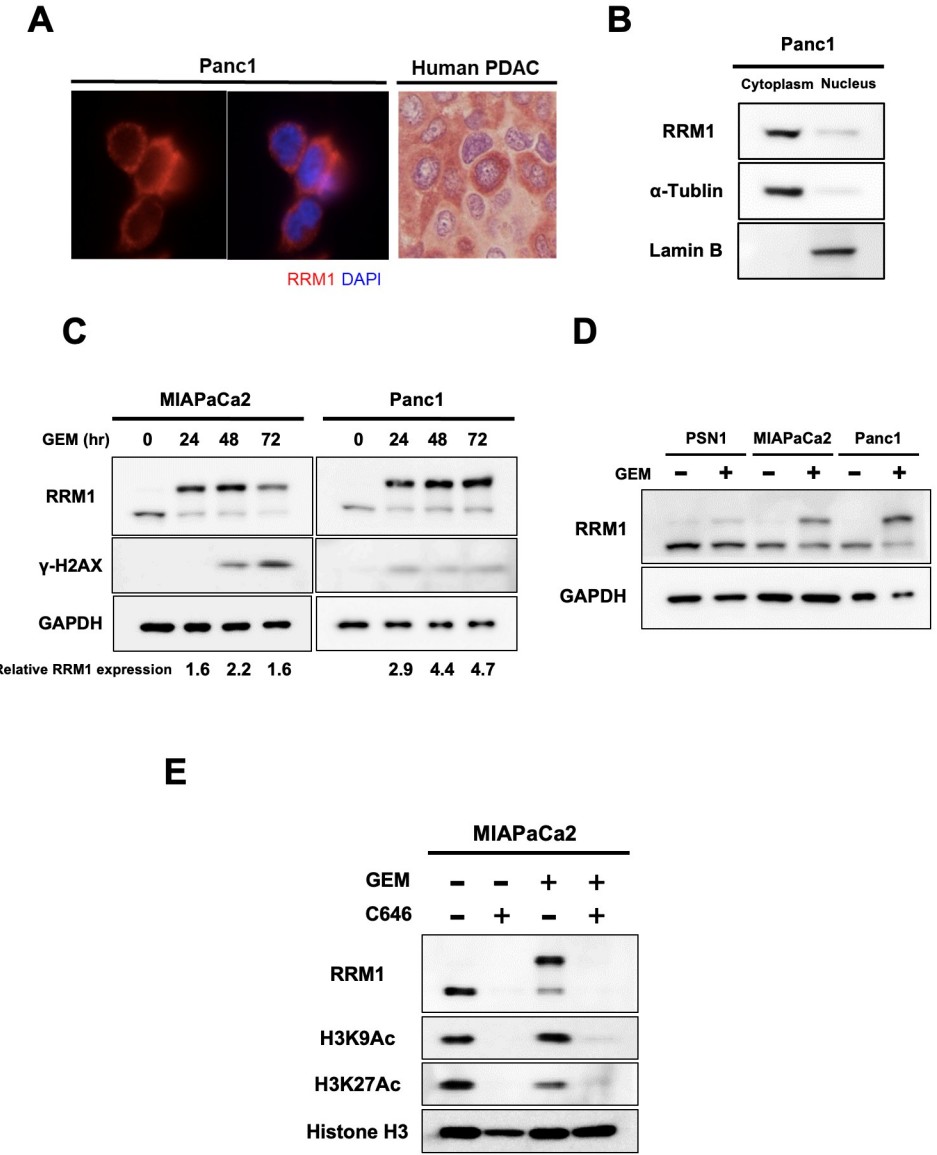

**Fig 4. Alteration of RRM1 expression after gemcitabine exposure in pancreatic cancer cells.** (**A**) Representative immunofluorescence of Panc1 cells and immunohistochemical staining of human pancreatic cancer cells for RRM1 expression showing that RRM1 is mainly localized in the cytoplasm. (**B**) Distribution of RRM1 in Panc1 cells. Proteins were separated into nuclear and cytoplasm fractions and examined by Western blotting. RRM1 was confirmed to be located in the cytoplasm. (**C**) Expression levels of RRM1 induced by gemcitabine from 24 hours to 72 hours. DNA damage was evaluated by γ-H2AX expression. Cells were treated with gemcitabine at 100 nM for MIAPaCa2 and 1 μM for Panc1 cells. RRM1 expression change with upper band shifting was observed at 24 hours after gemcitabine exposure. The numbers at the bottom indicate the relative intensity of the bands for upper band shifted RRM1 expression after gemcitabine exposure comparting to corresponding controls of original RRM1 intensity with no gemcitabine treatment (normalized by GAPDH). (**D**) Effects of gemcitabine treatment on RRM1 expression level in PSN1, MIAPaCa2, and Panc1 cells. The cells were treated with gemcitabine for 72 hours (at 10 nM for PSN1, 20 nM for MIAPaCa2, and 1 μM for Panc1). The doses of gemcitabine for each cell line were selected to have the sufficient cytotoxic effect of gemcitabine (S2B Fig). Gemcitabine exposure induced band shifted RRM1 alteration in pancreatic cancer cells. (**E**) Effects of C646 on expression levels of RRM1. Cells were pretreated with C646 at 50 μM for 24 hours and then treated additionally with gemcitabine at 25 nM for 24 hours. Acetylation status was evaluated by H3K27 and H3K9 expression using Western blotting. C646 decreased original RRM1 expression and band shifted RRM1 expression induced by gemcitabine.

**A**

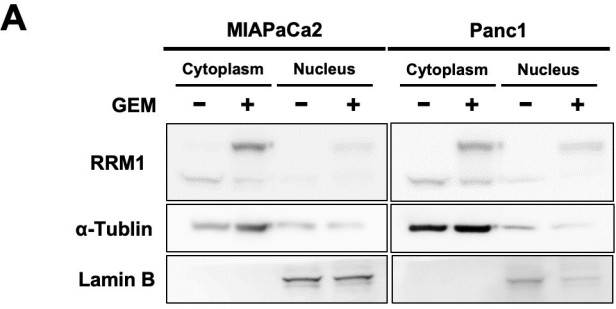

**B**

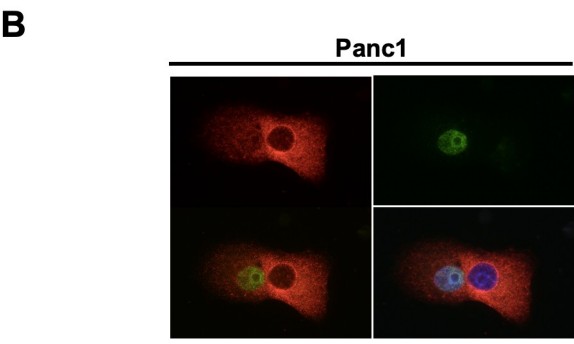

**C**

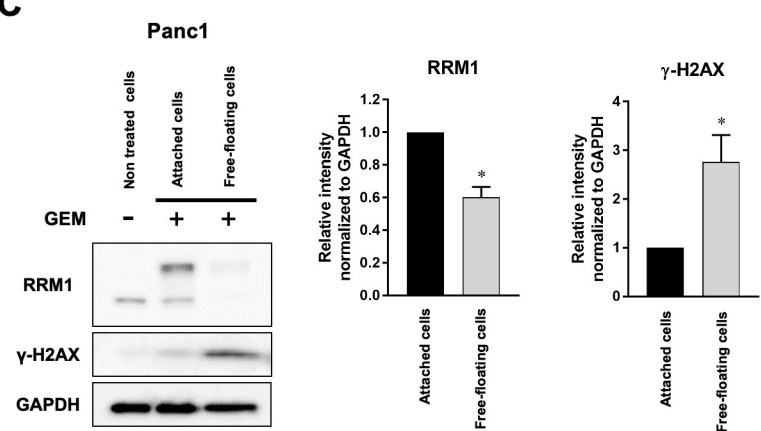

**Fig 5. Increased RRM1 expression in pancreatic cancer cell cytoplasm is associated with drug resistance to gemcitabine. (A)** Effects of gemcitabine on RRM1 expression levels in cytoplasmic and nuclear fraction in MIAPaCa2 and Panc1 cells. RRM1 activation induced by gemcitabine was predominantly observed in the cytoplasm. **(B)** Evaluation of RRM1 expression on exposure to gemcitabine. Notably, RRM1 activation in the cytoplasm was confirmed in those cells tolerating gemcitabine exposure. In contrast, RRM1 activation was not observed in cells with severe DNA damage induced by gemcitabine. **(C)** Cancer cells were fractionated into either still attached on the plate or free-floating in the medium after gemcitabine treatment for 72 hours. The latter showed marked γ-H2AX activation. Indeed, RRM1 activation in the cytoplasm was confirmed only in attached and viable cells to tolerate gemcitabine-induced DNA damage. On the other hand, cancer cells undergoing apoptosis increased DNA damage and significantly downregulated RRM1 expression. The graph depicts averaged intensity of bands representing γ-H2AX and band shifted RRM1 comparing to original RRM1 intensity without gemcitabine, normalized to the intensity of bands representing GAPDH. Error bars represent mean ± SD. *$p < 0.05$ vs attached viable cells after gemcitabine exposure. The Western blotting assay was performed in triplicate.

Of particular interest, RRM1 expression was significantly associated with worse postoperative overall survival rather than disease free survival for the patients with adjuvant

chemotherapy as shown in Fig 2A and 2B. Combination chemotherapies such as gemcitabine with nab-paclitaxel and FOLFIRINOX were currently performed for the treatment of pancreatic cancer in the recurrent or metastatic settings. Gemcitabine and 5-FU, these DNA targeting anticancer drugs play a central role in the treatment in the post relapse settings. RRM1 may exert its effect on resistance to multidrug therapy, including DNA-targeted anticancer drugs using after relapse relatively for a long time. Therefore, RRM1 expression probably affects overall survival than disease free survival in pancreatic cancer patients with adjuvant chemotherapy.

In this study, we attempted to figure out the biological function of RRM1 against gemcitabine. We observed synergistic effects of combining gemcitabine and RRM1 inhibition using specific RRM1 siRNA, or hydroxyurea (Fig 3B and 3C). The synergistic effect of RRM1 inhibition can be also expected for S-1. Supporting our results, other reports indicated that RRM1 is associated with therapeutic sensitivity of gemcitabine and fluorouracil in cancer cells [9,31]. Our results identified that the RRM1 expression status of the individual tumor may be useful for the determination of therapeutic efficacy as indicated in Fig 2C and 2D. In addition, gemcitabine with additional RRM1 inhibition could be a powerful treatment option as an adjuvant therapy especially for the patients with high RRM1 expression.

The relevance of RRM1 expression to gemcitabine resistance has been reported. RRM1 expression increases as tumor cells acquire resistance against chemotherapeutic agents including gemcitabine. However, the functional role of RRM1 was not understood, particularly in the acute response to gemcitabine exposure. In our study, we focused on cellular localization of RRM1 after exposure to this drug. Recently, ribonuclease reductase has been shown to accumulate at DNA damage sites in mammalian cells to facilitate localization of dNTP supplies for damage repair [32]. Niida and colleagues described RRM1 recruitment to nuclear damage foci colocalizing with γ-H2AX after irradiation [32]. In our model of gemcitabine exposure of pancreatic cancer cells, drug-induced DNA damage activated RRM1 expression in the cytoplasm (Fig 5A). RRM1 was not translocated to the nucleus. Notably, cytoplasmic RRM1 activation was observed only in cancer cells overcoming gemcitabine-induced DNA damage, implying that it is part of a cellular recovery process following exposure to the drug (Fig 5B and 5C). In contrast, this was not seen in cancer cells with severely damaged DNA. The biological significance of cytoplasmic RRM1 activation following exposure to gemcitabine is inextricably linked with its exertion of biological signaling resulting in acquisition of a resistant phenotype rather than participating in DNA repair at damage foci.

Minami and colleagues reported that knockdown of RRM1 increases intracellular gemcitabine accumulation by increasing the expression of uptake transporters [18]. Nakano and colleagues demonstrated that the ratio of hENT1-dCK to RRM1-RRM2 gene expression significantly correlated with gemcitabine sensitivity and decreased the development of drug resistance [16]. Concerning the regulation of RRM1 expression, this has remained unclear thus far and we hypothesize that epigenetic modification might be associated with the acute response to gemcitabine. Evidence has been accumulating, including from the present study, that histone acetylation is involved in DNA damage repair signaling [26,32]. Indeed, an increase in histone acetylation occurs at damage sites, coordinating with DNA damage repair complexes [32]. This evidence also supports our previous study in the model of gemcitabine exposure [26]. In the current study, RRM1 activation is regulated presumably in part through histone acetylation. Otherwise, RRM1 activation is associated with kinase activities, such as phosphorylation, even though the role of RRM1 phosphorylation remains uncertain for the repair of DNA damage caused by gemcitabine. Further exploration is necessary to clarify the mechanisms regulating RRM1 activation in the acute response to gemcitabine exposure.

In summary, we demonstrated that RRM1 expression is significantly associated with poorer postoperative prognosis of pancreatic cancer patients. Our study documented that cytoplasmic RRM1 activation is an important hallmark of cancer cells acquiring gemcitabine resistance in the acute response to drug exposure. For those pancreatic cancer patients with high RRM1 expression in their tumors, adjuvant chemotherapy with gemcitabine together with RRM1 inhibition might be an attractive treatment strategy leading to better postoperative outcomes.

## Supporting information

**S1 Fig. Immunohistochemical staining of total 12 cases of low-grade malignant pancreatic diseases. (A)** Serous cyst neoplasia (SCN), **(B)** mucinous cyst neoplasia (MCN), **(C)** intraductal papillary mucinous neoplasia (IPMN) with low-grade dysplasia, and **(D)** IPMN with intermediate-grade dysplasia. No RRM1 expression was detected in all benign cases without malignant potential.
(TIF)

**S2 Fig. Dose-response curve for hydroxyurea and gemcitabine. (A), (B)** Cells ($5$–$7.5 \times 10^3$ per well) were seeded and incubated overnight. Then each concentration of hydroxyurea or gemcitabine was administrated for 72 hours. **(A)** Cell viability analysis following hydroxyurea treatment in PSN1 and MIAPaCa2 cells (10 nM-10 mM). IC50 concentration for hydroxyurea was calculated at 140 μM for PSN1 and 328 μM for MIAPaCa2 cells. Cell viability was performed by WST-8 assay. **(B)** Cell viability analysis following gemcitabine treatment in PSN1, MIAPaCa2, and Panc1 cells (PSN1 and MIAPaCa2; 0.1 nM-1 μM, Panc1; 0.1 nM-10 μM).
(TIF)

**S3 Fig. RRM1 alteration level after gemcitabine exposure in MIAPaCa2 and PSN1.** Cancer cells were treated with siRRM1 or siNC for 12 hours. Subsequently gemcitabine was treated for 48 hours (MIAPaCa2; 20 nM and PSN1; 7.5nM). Top, representative Western blotting; bottom, the graph depicts averaged intensity of bands representing band shifted RRM1 expression comparing to original RRM1 intensity without gemcitabine, normalized to the intensity of bands representing GAPDH. Error bars represent mean ± SD. *p<0.05 vs cells with gemcitabine treated MIAPaCa2 cells by unpaired t-test. The Western blotting assay was performed in triplicate.
(TIF)

**S4 Fig. Correlation between the level of expression of RRM1 and postoperative outcomes in patients who received GEM and S-1 adjuvant chemotherapy. (A)** Kaplan-Meier curve for overall survival of patients who received S-1 adjuvant chemotherapy (S-1 group). **(B)** Kaplan-Meier curve for overall survival of patients who received gemcitabine-based adjuvant chemotherapy (GEM group).
(TIF)

**S1 Raw images.**
(PDF)

## Author Contributions

**Conceptualization:** Hiroaki Ono, Shinji Tanaka.

**Data curation:** Tomotaka Kato, Hiroaki Ono.

**Formal analysis:** Tomotaka Kato, Mikiya Fujii.

**Funding acquisition:** Hiroaki Ono, Minoru Tanabe.

**Investigation:** Tomotaka Kato, Hiroaki Ono, Mikiya Fujii.

**Methodology:** Hiroaki Ono.

**Project administration:** Hiroaki Ono.

**Resources:** Keiichi Akahoshi, Toshiro Ogura, Kosuke Ogawa, Daisuke Ban, Atsushi Kudo, Shinji Tanaka, Minoru Tanabe.

**Supervision:** Minoru Tanabe.

**Validation:** Hiroaki Ono.

**Visualization:** Tomotaka Kato, Hiroaki Ono.

**Writing – original draft:** Tomotaka Kato.

**Writing – review & editing:** Hiroaki Ono, Shinji Tanaka, Minoru Tanabe.

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
