## [Decision Letter · Decision Letter 0]

12 Nov 2020

PONE-D-20-25361

Cytoplasmic RRM1 activation as an acute response to gemcitabine treatment is involved in acquired resistance of pancreatic cancer cells

PLOS ONE

Dear Dr. Ono,

Thank you for submitting your manuscript to PLOS ONE. After careful consideration, we feel that it has merit but does not fully meet PLOS ONE’s publication criteria as it currently stands. Therefore, we invite you to submit a revised version of the manuscript that addresses the points raised by both reviewers during the review process.

We look forward to receiving your revised manuscript.

Kind regards,

Irina V. Lebedeva, Ph.D.

Academic Editor

PLOS ONE

Journal Requirements:

2)  Thank you for including your ethics statement:  "The use of archived specimens for this study was approved by the Ethics Committees of the Faculty of Medicine of Tokyo Medical and Dental University (permission No. M2000-1080, G2017-018).".   

Please provide additional details regarding participant consent. In the ethics statement in the Methods and online submission information, please ensure that you have specified (1) whether consent was informed and (2) what type you obtained (for instance, written or verbal, and if verbal, how it was documented and witnessed). If your study included minors, state whether you obtained consent from parents or guardians. If the need for consent was waived by the ethics committee, please include this information.

3) Please include your tables as part of your main manuscript and remove the individual files. Please note that supplementary tables (should remain/ be uploaded) as separate "supporting information" files

4) Please include captions for your Supporting Information files at the end of your manuscript, and update any in-text citations to match accordingly. Please see our Supporting Information guidelines for more information: http://journals.plos.org/plosone/s/supporting-information.

5) We note that you have included the phrase “data not shown” in your manuscript. Unfortunately, this does not meet our data sharing requirements. PLOS does not permit references to inaccessible data. We require that authors provide all relevant data within the paper, Supporting Information files, or in an acceptable, public repository. Please add a citation to support this phrase or upload the data that corresponds with these findings to a stable repository (such as Figshare or Dryad) and provide and URLs, DOIs, or accession numbers that may be used to access these data. Or, if the data are not a core part of the research being presented in your study, we ask that you remove the phrase that refers to these data.

6) PLOS ONE now requires that authors provide the original uncropped and unadjusted images underlying all blot or gel results reported in a submission’s figures or Supporting Information files. This policy and the journal’s other requirements for blot/gel reporting and figure preparation are described in detail at https://journals.plos.org/plosone/s/figures#loc-blot-and-gel-reporting-requirements and https://journals.plos.org/plosone/s/figures#loc-preparing-figures-from-image-files. When you submit your revised manuscript, please ensure that your figures adhere fully to these guidelines and provide the original underlying images for all blot or gel data reported in your submission. See the following link for instructions on providing the original image data: https://journals.plos.org/plosone/s/figures#loc-original-images-for-blots-and-gels.

Reviewers' comments:

Reviewer's Responses to Questions

**Comments to the Author**

1. Is the manuscript technically sound, and do the data support the conclusions?

Reviewer #1: Yes

Reviewer #2: Partly

2. Has the statistical analysis been performed appropriately and rigorously? 

Reviewer #1: Yes

Reviewer #2: Yes

3. Have the authors made all data underlying the findings in their manuscript fully available?

Reviewer #1: Yes

Reviewer #2: Yes

4. Is the manuscript presented in an intelligible fashion and written in standard English?

Reviewer #1: Yes

Reviewer #2: Yes

5. Review Comments to the Author

Reviewer #1: The manuscript reports the investigation of RRM1 in pancreatic cancer prognosis as well as begins to identify cellular responses associated with efficacy of gemcitabine therapy. The following comments are offered:

1. The role of the expression in RRM1 in survival of pancreatic cancer is less controversial than they present. Please see publications by Sierzega et al, Pancreas 2017, and the meta-analysis of Wei et al, Pancreas 2013.

2. The authors refer to RRM1 activation as part of the acquired resistance to gemcitabine. This is not the proper use of the concept. Acquired resistance would be evolution or selection of a subclone that is resistant to gemcitabine after exposure. Their investigation reports inherent or de novo resistance.

3. The authors need to provide information on the quantification of the IHC and how they set cutoff points for "high" and "low" expression.

4. The data related to RRM1 expression and resistance to chemotherapy (Figure 2C, C) would be better presented if they compared the RRM1 high group treated with and without gemcitabine, anticipating no benefit of gemcitabine, and comparing the RRM1 low group treated with and without gemcitabine in which a benefit was observed.

5. The Results section of the manuscript around the data of Figure 3 is very confusing. They refer to Figure 3B in the text, but it seems like they are referring to Figure 3C (Figure 3B is the reduced expression of RRM1 after siRNA treatment). In addition, there are no Figure legends for Figures 3D and 3E.

6. The data presented in Figure 3E is difficult to interpret. How does the data show increased DNA damage accumulation.

7. The investigation that cytoplasmic activation of RRM1 is related to the baseline levels of RRM1? Namely for those cells with high baseline, do they demonstrate the cytoplasmic activation to the same extent as those with low RRM1? The data of Figure 3A shows some variation in baseline RRM1 in which HS766T may be worth investigating.

Reviewer #2: The authors evaluate the clinical significance of RRM1 expression in gemcitabine resistance in pancreas cancer. Drug resistance in pancreas cancer is a major problem. DNA damage resistance has been touted as a primary mechanism for GEM resistance. Literature is confusing as to which DNA damage repair regulators including RRM1, RRM2, EMCC1, and others, or their collaboration are important. This manuscript focuses specifically on RRM1. My concerns including key ones (#9 and 11) are below.

Protein expression in primary tissue by IHC is subjected to interpretation by pathologists. Please add detail in the method for who did the grading and whether there was more than one pathologist. Are they reviewing the staining together or separately and how they rectified varying expression in the same tissue and if the pathologists do not agree on the staining intensity.

Pancreas cancer treatment has improved, so a specific period between the first and last cases need to be included.

Please clarify the non-GEM group for what adjuvants these patients received. It is important to know whether they received DNA targeting agents since high RRM1 seems to also be associated with poor survival in this group. Even though there is a trend towards worse survival in the GEM group, the data is not statistically significant. Need to discuss this. Could be a limited sample size but the negative data here does not support further studies of RRM1 and GEM resistance.

Please discuss the negative PFS with adjuvant therapy but the highly significant OS. This data would suggest an adverse impact of RRM1 in the post relapse setting, unrelated to GEM

Please describe the siRRM1 cells whether they have different growth rates. This will matter more in your figure 1 when you compare the GEM treatment effects.

In Figure 3A: please do densitometric analysis and describe the ratio between RRM1 and GAPDH. The MIAPaCa cells look to have the lowest amount of RRM1, so the role of RRM1 is better studied in the overexpression model. PSN is a good model here

Figure 3B and C, please add data of siRMM cells without GEM. That will show whether there are any baseline cumulative DNA damage and different growth rate

Figure 3D: Need HU treatment alone as a control too since HU dose chosen was based on IC50, per author’s description (line 233)

Fig 4C showing altered MW of RRM1 with GEM treatment is interesting but C646 decreased both primary and higher MW forms, so cannot use this method to conclude that the altered MW is via acetylation and cannot conclude that acetylation of RRM1 has any significant role in DNA damage. To support this, it would need evidence of a different degree of DNA damage and cell death in GEM +/- C646. The posttranslational modification of RRM1 is an interesting one esp. if it affects the localization and function of the protein. This is a novel finding and needs more expansion of the experiment and/or discussion.

Please do densitometry for the ration of cytoplasmic and nuclear RRM1 in Fig 4E. looks like there is an increase in a nuclear fraction too

The description of accumulation of cytoplasmic RRM1 only in cells without DNA damage could be interesting but this needs quantification, rather than just one immunofluorescent staining and should be compared between GEM sensitive and resistant cell line. This point is critical as abnormal translocation of RRM1 and its effect on GEM sensitivity is a novel finding while other clinical roles of RRM1 and its function as determined by siRNA in cell lines have already been previously reported.

6. PLOS authors have the option to publish the peer review history of their article (what does this mean?). If published, this will include your full peer review and any attached files.

Reviewer #1: No

Reviewer #2: No

---

## [Author Response · Author response to Decision Letter 0]

26 Dec 2020

Irina V. Lebedeva, Ph.D.

Academic Editor

PLOS ONE

Dear Dr. Lebedeva,

We would like to thank all the reviewers of our manuscript for their insightful comments and helpful suggestions. We have added new data and revised the manuscript accordingly. Each of the issues raised by the reviewers is addressed specifically below.

Reviewer #1:

1. The role of the expression in RRM1 in survival of pancreatic cancer is less controversial than they present. Please see publications by Sierzega et al, Pancreas 2017, and the meta-analysis of Wei et al, Pancreas 2013.

Comments: Thank you for your valuable opinion. We agree with reviewer’s comments about the description for the clinical significance of RRM1 expression. The reports by Wei et al (Pancreas 2013) and Sierzega et al (Pancreas 2017) were referred to in our manuscript and we revised our statement in the introduction section described as below．

“RRM1 has also been reported to affect disease prognosis of pancreatic cancer. In fact, the meta-analysis for RRM1 demonstrated that high expression of RRM1 was significantly associated with worse overall survival. The clinical impact of RRM1 has been elucidated. However, the functions of RRM1 in pancreatic cancer biology remain unclear.”

2. The authors refer to RRM1 activation as part of the acquired resistance to gemcitabine. This is not the proper use of the concept. Acquired resistance would be evolution or selection of a subclone that is resistant to gemcitabine after exposure. Their investigation reports inherent or de novo resistance.

Comments: Thank you for your important advice. As the reviewer pointed out, we have modified acquired resistance to drug resistance in our title and manuscript. 

3. The authors need to provide information on the quantification of the IHC and how they set cutoff points for "high" and "low" expression.

Comments: According to reviewer’s suggestion, we revised our statement in the Figure Legends section to clarify the definitions for staining of RRM1 expression. We added the following sentence in the Figure Legends section.

“For the immunohistochemical staining of RRM1 in pancreatic cancer, the islets of Langerhans served as a positive internal control to evaluate the expression. RRM1 expression was graded into RRM1 expression levels as low (no staining or weak intensity staining comparing to internal control in less than 30% of cells) or high (strong intensity staining in more than 30% of cells based on cytoplasmic staining intensity). The staining grade of RRM1 expression was assessed by two investigators and reviewed by one pathologist.”

4. The data related to RRM1 expression and resistance to chemotherapy (Figure 2C, C) would be better presented if they compared the RRM1 high group treated with and without gemcitabine, anticipating no benefit of gemcitabine, and comparing the RRM1 low group treated with and without gemcitabine in which a benefit was observed.

Comments: We agree with the reviewer’s suggestion. As reviewer pointed out, we revised the data related to RRM1 expression and resistance to chemotherapy in Figure 2C and 2D. Figure 2C and 2D represent the clinical efficacy of RRM1 expression on a benefit of adjuvant chemotherapy comparing low and high RRM1 expression in the revised manuscript. Thanks to the reviewer’s valuable advice, our assertion concerning the biological role of RRM1 on clinical benefit of adjuvant therapy has been clarified. 

5. The Results section of the manuscript around the data of Figure 3 is very confusing. They refer to Figure 3B in the text, but it seems like they are referring to Figure 3C (Figure 3B is the reduced expression of RRM1 after siRNA treatment). In addition, there are no Figure legends for Figures 3D and 3E.

Comments: We appreciate the reviewer’s meticulous review in the very detail point. The Result section of Figure 3B and 3C and corresponding Figure legends have been changed appropriately. And we have added Figure legends for Figure 3D and 3E in our revised manuscript. We apologize for this careless mistake in our figures and figure legends and have corrected this in the revision. 

6. The data presented in Figure 3E is difficult to interpret. How does the data show increased DNA damage accumulation.

Comments: According to reviewer’s suggestion, the experiment was re-performed for western blotting to clarify increased DNA damage accumulation in MIAPaCa2 and PSN1 cells after hydroxyurea and gemcitabine treatment.

7. The investigation that cytoplasmic activation of RRM1 is related to the baseline levels of RRM1? Namely for those cells with high baseline, do they demonstrate the cytoplasmic activation to the same extent as those with low RRM1? The data of Figure 3A shows some variation in baseline RRM1 in which HS766T may be worth investigating.

Comments: We really appreciate for your raising an important question and providing these insights. Since it is especially important on this point in the manuscript, we have performed some additional experiments. In our opinion, cytoplasmic activation of RRM1 is not related to the baseline level of RRM1 expression. We rather speculated that this activation is related in response to DNA damage repair in pancreatic cancer cells. In fact, the basal RRM1 expression in PSN1 cells was higher than MIAPaCa2 cells as shown in Figure 3A. Whereas, RRM1 alteration after gemcitabine exposure was significantly scant in PSN1 cells, which are gemcitabine-sensitive comparing to MIAPaCa2 cells as shown in the newly added Supplementary Figure S3 and Figure 4. In addition, cross-sectional analysis of pancreatic cancer cell lines has been added to the revised manuscript as shown Figure 4D. Among pancreatic cancer cell lines we tested, Panc1 cells exhibited the most resistant phenotype to gemcitabine (Supplementary Figure S3). The results of these experiments have supported our notion very well indeed. These results are added in Figure 4D and Supplementary Figure S3 and S4. The relevant parts of Results and Figure Legends sections have been revised accordingly.

Reviewer #2: 

#1 Protein expression in primary tissue by IHC is subjected to interpretation by pathologists. Please add detail in the method for who did the grading and whether there was more than one pathologist. Are they reviewing the staining together or separately and how they rectified varying expression in the same tissue and if the pathologists do not agree on the staining intensity.

Comments: Thank you for your valuable suggestion. In this study, two investigators assessed the staining grade of RRM1 expression, and one pathologist judged and reviewed the results separately. For the immunohistochemical staining of RRM1 in pancreatic cancer, the islets of Langerhans served as a positive internal control to evaluate the expression. The grading of RRM1 protein expression in primary tissues was determined into two grades. Tumor cells which had stronger intensity staining comparing to positive control in more 30% of cells were defined as high RRM1 expression. Whereas low RRM1 expression was defined as no staining or weak intensity staining in less 30% of cells. The staining grade of RRM1 expression was assessed by two investigators and the pathologist agreed with determination of the grade and the criteria of RRM1 expression. We revised our statement in the Figure Legends section to clarify the definitions for staining of RRM1 expression.

#2 Pancreas cancer treatment has improved, so a specific period between the first and last cases need to be included.

Comments: We agree with reviewer’s opinion. We revised our statement with a specific period and relevant parts of the Material and Methods were revised accordingly.

#3 Please clarify the non-GEM group for what adjuvants these patients received. It is important to know whether they received DNA targeting agents since high RRM1 seems to also be associated with poor survival in this group. Even though there is a trend towards worse survival in the GEM group, the data is not statistically significant. Need to discuss this. Could be a limited sample size but the negative data here does not support further studies of RRM1 and GEM resistance.

Comments: In non-GEM group, S-1 was used as a therapeutic agent for the patients in the postoperative adjuvant setting. S-1 is an oral fluoropyrimidine compound comprising tegafur, gimeracil, and oteracil potassium. S-1 functions as a prodrug of 5-FU and, like gemcitabine, is classified as a DNA targeting agent. The therapeutic use of S-1 as postoperative adjuvant chemotherapy has become the standard of care in Japan. One of the reasons why there was no significant difference in the GEM adjuvant group is the issue about the limited sample size. Another reason is that the overall prognosis in the GEM group was worse than that in the S-1 group, which may indicate that GEM was less effective than S-1 as an adjuvant treatment in addition to the different situation background. In this study, we focused on the biological function of RRM1 against gemcitabine, but the synergistic effect of RRM1 can be expected for S-1, which is also characterized as a DNA-targeted therapeutic agent. The phrase “non-GEM group” was misleading and have corrected. The details about the difference of therapeutic efficacy between gemcitabine and S-1 was described in the Discussion section and the figures has been changed to supplemental materials (Supplemental Figure S2A and S2B). We have accordingly revised the relevant parts of the Results, Figure Legends, and Discussion sections.

#4 Please discuss the negative PFS with adjuvant therapy but the highly significant OS. This data would suggest an adverse impact of RRM1 in the post relapse setting, unrelated to GEM.

Comments: It has been already common knowledge that GnP (gemcitabine and nab-paclitaxel) or FOLFIRINOX (5-FU, oxaliplatin, and irinotecan) were currently performed for the treatment of pancreatic cancer in the recurrent or metastatic settings. DNA targeting anticancer drugs such as gemcitabine and 5-FU play a central role in the treatment even in the post relapse settings. RRM1 may exert its effect on resistance to multidrug therapy, including DNA-targeted anticancer drugs using after relapse relatively for a long time. Therefore, RRM1 expression probably affects OS rather than DFS in pancreatic cancer patients. A description of these contents has been added to the Discussion section.

#5 Please describe the siRRM1 cells whether they have different growth rates. This will matter more in your figure 1 when you compare the GEM treatment effects.

Comments: We added the experimental data using siRRM1 cells without gemcitabine exposure in Figure 3B. Cell proliferation was significantly inhibited in siRRM1 cells as well, regardless of basal RRM1 expression. We have accordingly revised the relevant parts of the Results and Figure Legends sections.

#6 In Figure 3A: please do densitometric analysis and describe the ratio between RRM1 and GAPDH. The MIAPaCa cells look to have the lowest amount of RRM1, so the role of RRM1 is better studied in the overexpression model. PSN is a good model here.

Comments: According to reviewer’s suggestion, we added the data of endogenous RRM1 expression in the four pancreatic cancer cell lines using densitometric analysis as the relative ratio comparing to MIAPaCa2 cells, which possessed the lowest RRM1 expression in our panel as shown in Figure 3A. The relevant parts of the Results and Figure Legends sections have been revised accordingly.

#7 Figure 3B and C, please add data of siRMM cells without GEM. That will show whether there are any baseline cumulative DNA damage and different growth rate

Comments: As reviewer’s indication, we added the data using siRRM1 cells without gemcitabine exposure to Figure 3B and 3D. Gene-silencing of RRM1 induced DNA damage accumulation and inhibited cell proliferation. These effects against DNA damage accumulation and inhibition of cell viability were synergistically enhanced when gemcitabine was used in combination. The relevant parts of the Results and Figure Legends sections have been revised accordingly.

#8 Figure 3D: Need HU treatment alone as a control too since HU dose chosen was based on IC50, per author’s description (line 233)

Comments: As reviewer’s indication, we added the data treating with HU alone to Figure 3C and 3E. Like RRM1 gene-silencing, HU monotherapy induced DNA damage accumulation and inhibited cell proliferation. These effects against DNA damage accumulation and inhibition of cell viability were also synergistically enhanced when gemcitabine was used in combination. The relevant parts of the Results and Figure Legends sections have been revised accordingly. 

#9 Fig 4C（D?）showing altered MW of RRM1 with GEM treatment is interesting but C646 decreased both primary and higher MW forms, so cannot use this method to conclude that the altered MW is via acetylation and cannot conclude that acetylation of RRM1 has any significant role in DNA damage. To support this, it would need evidence of a different degree of DNA damage and cell death in GEM +/- C646. The posttranslational modification of RRM1 is an interesting one esp. if it affects the localization and function of the protein. This is a novel finding and needs more expansion of the experiment and/or discussion.

Comments: Previously we reported that a histone acetylation inhibitor C646 increased gemcitabine-induced DNA damage and decreased cell viability in pancreatic cancer cells (Ono H et al, Oncotarget, 2016; 7:51301-5131). Indeed, it is difficult to show a direct relevance between RRM1 acetylation and DNA damage using C646. Since C646 globally represses the transcriptional regulation mediated by downregulation of histone acetylation, therefore it is not suitable for evaluation of simply repression of RRM1 acetylation. We demonstrated that using densitometric analysis, altered molecular weight of RRM1 was significantly inhibited by a histone acetylation inhibitor C646 instead. We agree that the posttranslational modification of RRM1 is an interesting point, but we have not been able to demonstrate the direct relevance of RRM1 acetylation on gemcitabine-induced DNA damage in the revised manuscript. We will leave it as a future research topic. The result was added in Figure 4E in the revised manuscript. The relevant parts of the Results and Figure Legends sections have been revised accordingly. 

 #10 Please do densitometry for the ration of cytoplasmic and nuclear RRM1 in Fig 4E. looks like there is an increase in a nuclear fraction too

Comments: As the reviewer mentioned, it appeared that there is also an increase in a nuclear fraction in Panc1 cells. We examined RRM1 expression in the nuclear and cytoplasmic fractions with another cancer cell line using MIAPaCa2 cells. As shown in Figure 5A, in MIAPaCa2 cells, there is a very slight increase in RRM1 expression in the nucleus. Most of band shifted RRM1 alteration occurred in the cytoplasm, whereas the amount of increase in RRM1 nuclear fraction is exceedingly small. At the same time, the band of α-Tublin can be recognized even in the nuclear fraction. Considering these results, we supposed that a partial inclusion of cytoplasmic components is observed due to the limitations of experimental precision rather than an increase in nuclear fractions. A simple comparative analysis of nuclear and cytoplasmic proteins is difficult, so we did not perform a densitometric analysis in this case. The results were added in Figure 5A in the revised manuscript. The relevant parts of the Results and Figure Legends sections have been revised accordingly. 

#11 The description of accumulation of cytoplasmic RRM1 only in cells without DNA damage could be interesting but this needs quantification, rather than just one immunofluorescent staining and should be compared between GEM sensitive and resistant cell line. This point is critical as abnormal translocation of RRM1 and its effect on GEM sensitivity is a novel finding while other clinical roles of RRM1 and its function as determined by siRNA in cell lines have already been previously reported.

Comments: Thank you for your valuable opinion, we completely agree with reviewer’s comments about the necessity of quantification to calculate accumulation of cytoplasmic RRM1 in response to DNA damage. We performed additional investigation for western blotting and subsequent densitometric analysis to examine the RRM1 expression level quantitatively in response to DNA damage by comparing attached on the plate and free-floating cancer cells. As shown in Figure 5C, the quantitative analysis demonstrated that apoptosis-inducing cancer cells abolished RRM1 activation and increased DNA damage accumulation. On the other hand, RRM1 activation in the cytoplasm was confirmed only in attached and viable cells to tolerate gemcitabine-induced DNA damage. We convince that this result quantitatively confirmed the finding that cytoplasmic RRM1 accumulated only in cells without DNA damage. The result was added in Figure 5C and the relevant parts of Results and Figure Legends sections have been revised accordingly. 

In conclusion, we would like to thank the reviewers for their careful reading of the manuscript and helpful suggestions. We believe the manuscript is much improved.

Sincerely, 

Tomotaka Kato, MD

Hiroaki Ono, MD PhD

---

## [Decision Letter · Decision Letter 1]

2 Mar 2021

PONE-D-20-25361R1

Cytoplasmic RRM1 activation as an acute response to gemcitabine treatment is involved in drug resistance of pancreatic cancer cells

PLOS ONE

Dear Dr. Ono,

Thank you for submitting your manuscript to PLOS ONE. After careful consideration, we feel that it has merit but does not fully meet PLOS ONE’s publication criteria as it currently stands. Therefore, we invite you to submit a revised version of the manuscript that addresses the points raised during the review process.

Please address the concerns highlighted by the reviewer 3.

We look forward to receiving your revised manuscript.

Kind regards,

Irina V. Lebedeva, Ph.D.

Academic Editor

PLOS ONE

Journal Requirements:

Reviewers' comments:

Reviewer's Responses to Questions

**Comments to the Author**

1. If the authors have adequately addressed your comments raised in a previous round of review and you feel that this manuscript is now acceptable for publication, you may indicate that here to bypass the “Comments to the Author” section, enter your conflict of interest statement in the “Confidential to Editor” section, and submit your "Accept" recommendation.

Reviewer #2: All comments have been addressed

Reviewer #3: (No Response)

2. Is the manuscript technically sound, and do the data support the conclusions?

Reviewer #2: Yes

Reviewer #3: Partly

3. Has the statistical analysis been performed appropriately and rigorously? 

Reviewer #2: Yes

Reviewer #3: Yes

4. Have the authors made all data underlying the findings in their manuscript fully available?

Reviewer #2: Yes

Reviewer #3: Yes

5. Is the manuscript presented in an intelligible fashion and written in standard English?

Reviewer #2: Yes

Reviewer #3: Yes

6. Review Comments to the Author

Reviewer #2: (No Response)

Reviewer #3: This study highlighted the significance of RRM1 expression in pancreatic cancer treatment and attempted to shed some light on its chen-resistance mechanism. There are a few issues that require some attention.

1. For the univariate and multivariate analysis, adjuvant therapy should be one of the variables included in the model, especially the remaining study has shown that adjuvant treatment could have significant survival impact on patients with low RRM1 expression. Thus, adjuvant treatment could be a confounding factor. Thus, this must be included in the multivariate regression model regardless of the P value in the univariate analysis.

2. Regarding the RRM1 band shift, the data did not clearly support the conclusion regarding the role of histone acetylation. If histone acetylation is responsible directly or indirectly for the upward shift of RRM1, one would expect a stronger bottom band and weaker top band with C646 treatment (fig 4E). Instead, it seems that both bands were lighter. So, histone acetylation does not seem to be responsible for this post-translation modification of RRM1, but perhaps on degradation or expression. It's probably some other protein modifications unrelated to the histone acetylation. How about ubiquitination, parylation, or sumolyation, etc.

3. In their conclusion, the authors stated "RRM1 activation". I am not sure what "activation" did the author refer to. All we have seen was a band shift with no clear mechanism. How do we know the higher molecular weight RRM1 is the active form?

7. PLOS authors have the option to publish the peer review history of their article (what does this mean?). If published, this will include your full peer review and any attached files.

Reviewer #2: **Yes: **Attaya Suvannasankha

Reviewer #3: No

---

## [Author Response · Author response to Decision Letter 1]

31 Mar 2021

Irina V. Lebedeva, Ph.D.

Academic Editor

PLOS ONE

Dear Dr. Lebedeva,

We would like to thank all the reviewers of our manuscript for their insightful comments and helpful suggestions. We have added new data and revised the manuscript accordingly. Each of the issues raised by the reviewers is addressed specifically below.

Review Comments to the Author

Reviewer #3: 

1. For the univariate and multivariate analysis, adjuvant therapy should be one of the variables included in the model, especially the remaining study has shown that adjuvant treatment could have significant survival impact on patients with low RRM1 expression. Thus, adjuvant treatment could be a confounding factor. Thus, this must be included in the multivariate regression model regardless of the P value in the univariate analysis.

Comments: 

Thank you for your important advice. As the reviewer pointed out, we added the parameter of the adjuvant treatment in the multivariate analysis in Table 2. It revealed non-adjuvant chemotherapy was one of the independent prognostic factors for overall survival. We revised our manuscript in the Results section accordingly. 

2. Regarding the RRM1 band shift, the data did not clearly support the conclusion regarding the role of histone acetylation. If histone acetylation is responsible directly or indirectly for the upward shift of RRM1, one would expect a stronger bottom band and weaker top band with C646 treatment (fig 4E). Instead, it seems that both bands were lighter. So, histone acetylation does not seem to be responsible for this post-translation modification of RRM1, but perhaps on degradation or expression. It's probably some other protein modifications unrelated to the histone acetylation. How about ubiquitination, parylation, or sumolyation, etc.

Comments: 

Thank you for your valuable question. We have repeated the experiment in Figure 4E with the additional administration of C646 monotherapy to address the reviewer’s suggestion. Interestingly, C646 decreased the expression level of RRM1. Although gemcitabine increased the band shift of RRM1, both the band shift after gemcitabine administration and the original expression level of RRM1 were decreased by C646 treatment. These experimental results indicated that RRM1 expression levels and changes in band shifted RRM1 expression are regulated by histone acetylation. Therefore, the relationship between histone acetylation and alteration of RRM1 expression after gemcitabine administration has been well established, and in this manuscript, we focused only on histone acetylation. The other translational modifications such as ubiquitination, parylation, and sumolyation, which were pointed out by the reviewer, are of very interest, however, will be the subject of future experiments. And we have clarified the relationship between RRM1 expression and histone acetylation has been clarified in our other papers under submission. The result is added in Figure 4E. The relevant parts of Results and Figure Legends sections have been revised accordingly.

3. In their conclusion, the authors stated "RRM1 activation". I am not sure what "activation" did the author refer to. All we have seen was a band shift with no clear mechanism. How do we know the higher molecular weight RRM1 is the active form?

Comments: 

We appreciate your insightful suggestion. In this study, we confirmed that RRM1 expression in the cytoplasm was indeed increased in cancer cells acquiring gemcitabine resistance. The Western blot in Figure 4C showed that the band shifted RRM1 expression increased in response to gemcitabine treatment compared to the original band. Figure 5C also showed that the band shifted RRM1 expression was associated with the acquisition of gemcitabine resistance. We showed the results of fluorescence immunostaining in Figure 5B, suggesting that gemcitabine-resistant cancer cells were likely to have increased band shifted RRM1 expression mainly in the cytoplasm. Accordingly, total RRM1 expression combined with original band and band shifted form was activated. It is certainly difficult at this point to determine whether the band shifted RRM1 is in an activated form or not. However, from these experimental results representing Figure 4B and other figures, we can finally conclude that increased total RRM1 expression as RRM1 activation is associated in the acute phase of gemcitabine resistance, regardless of whether it is band shifted or original.

In consideration of the above, we have corrected the manuscript in the Abstract section, without including any definitive expressions that band shift RRM1 expression is an active form after gemcitabine exposure. 

In conclusion, we would like to thank the reviewers for their careful reading of the manuscript and helpful suggestions. We believe the manuscript is much improved.

Sincerely yours, 

Tomotaka Kato, MD

Hiroaki Ono, MD PhD

---

## [Decision Letter · Decision Letter 2]

26 May 2021

Cytoplasmic RRM1 activation as an acute response to gemcitabine treatment is involved in drug resistance of pancreatic cancer cells.

PONE-D-20-25361R2

Dear Dr. Ono,

We’re pleased to inform you that your manuscript has been judged scientifically suitable for publication and will be formally accepted for publication once it meets all outstanding technical requirements.

Kind regards,

Irina V. Lebedeva, Ph.D.

Academic Editor

PLOS ONE

Additional Editor Comments (optional):

Reviewers' comments:

Reviewer's Responses to Questions

**Comments to the Author**

1. If the authors have adequately addressed your comments raised in a previous round of review and you feel that this manuscript is now acceptable for publication, you may indicate that here to bypass the “Comments to the Author” section, enter your conflict of interest statement in the “Confidential to Editor” section, and submit your "Accept" recommendation.

Reviewer #3: All comments have been addressed

2. Is the manuscript technically sound, and do the data support the conclusions?

Reviewer #3: Yes

3. Has the statistical analysis been performed appropriately and rigorously? 

Reviewer #3: Yes

4. Have the authors made all data underlying the findings in their manuscript fully available?

Reviewer #3: Yes

5. Is the manuscript presented in an intelligible fashion and written in standard English?

Reviewer #3: Yes

6. Review Comments to the Author

Reviewer #3: The authors have appropriately addressed all the comments and modified their conclusion based on their findings and limitations of the data. No further comments.

7. PLOS authors have the option to publish the peer review history of their article (what does this mean?). If published, this will include your full peer review and any attached files.

Reviewer #3: No

---

## [Editor Report · Acceptance letter]

1 Jun 2021

PONE-D-20-25361R2 

Cytoplasmic RRM1 activation as an acute response to gemcitabine treatment is involved in drug resistance of pancreatic cancer cells. 

Dear Dr. Ono:

I'm pleased to inform you that your manuscript has been deemed suitable for publication in PLOS ONE. Congratulations! Your manuscript is now with our production department. 

Kind regards, 

on behalf of

Dr. Irina V. Lebedeva 

Academic Editor

PLOS ONE